

# Revisiting Kelvin Helmholtz Instabilities and von Kármán Vortices in Canopy Turbulence.

Tirtha Banerjee[1,2], Frederik De Roo[2], and Rodman Linn[1]

[1](Current) Earth and Environmental Sciences Division, Los Alamos National Laboratory, New Mexico 87545, USA
[2]Karlsruhe Institute of Technology (KIT), Institute of Meteorology and Climate Research, Atmospheric Environmental Research (IMK-IFU), 82467 Garmisch-Partenkirchen, Germany

*Correspondence to:* Tirtha Banerjee (tirtha.banerjee@lanl.gov, banerjeetirtha10@gmail.com)

**Abstract.** Studying turbulence in vegetation canopies is important in the context of a number of micrometeorological and hydrological applications. While recent focus has shifted more towards exploring different kinds of canopy heterogeneities, there are still gaps in the existing knowledge on the multiple types of dynamics involved in the case of horizontally homogeneous canopies. For example, experimental studies have indicated that turbulence in the canopy sublayer (CSL) can be divided into three regimes. In the deep-zone, the flow-field is dominated by von Kármán vortex streets and interrupted by strong sweep events. The second zone near the canopy top is dominated by attached eddies and Kelvin-Helmholtz waves associated with the velocity inflection point in the mean longitudinal velocity profile. Above the canopy, the flow resembles classical boundary layer flow. In this study, these different kinds of dynamics are studied together by means of a large eddy simulation (LES). The main theme of this work is to address the question whether the parametrization of the canopy by a distributed drag force in numerical simulations instead of placing real solid obstacles is consistent with the three layer conceptual model. Unique techniques such as measures from information theory and coupled oscillator analysis are used to extract the coherent structures associated with the two motions. It can be stated that a better understanding of the rich dynamics associated with the simplest case of canopy turbulence can lead to more efficient simulations and more importantly improve the interpretation of more complex scenarios.

## 1 Introduction

Understanding the role of turbulent flow in the canopy sub layer (CSL)- the region from the ground surface up to 2-5 times the canopy height (Raupach and Thom, 1981), has several practical implications. Transport and distribution of of scalars such as carbon dioxide, water vapor, trace gases, aerosols etc. in the CSL is determined by the generic feature of turbulence in that region, which consequently influences biosphere-atmosphere interaction. For example, a recent study by Fuentes et al. (2016) has shown how organized non-local motions such as sweeps and ejections control the mixing ratios and residence times of plant emitted hydrocarbons in tropical forests. The intensity of turbulence determines how much of the hydrocarbons undergo oxidation and convert to aerosols which are released into the overlying atmosphere and eventually create cloud condensation nuclei, influencing rainfall. This example demonstrates the importance of an improved understanding of CSL turbulent structures, which is an interesting research problem combining fluid mechanics with hydrology, ecology, atmospheric and climate





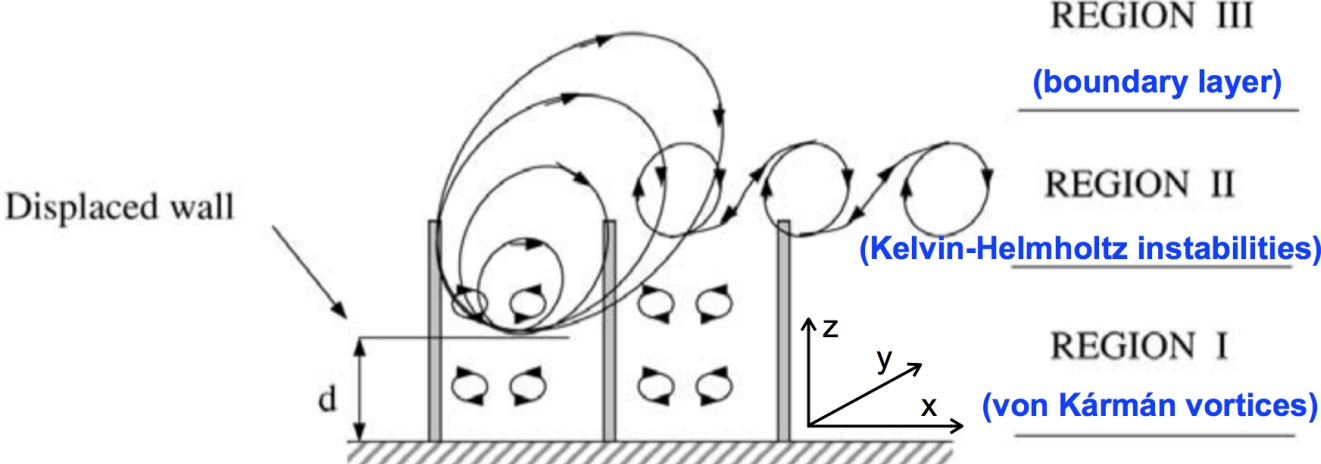

**Figure 1.** Conceptual model of canopy turbulence by Poggi et al. (2004b)

sciences. Other potential applications range from seed dispersion, air quality monitoring, land use strategy to studying effects of disturbances such as vegetation thinning, forest fires, insect outbreak etc. on the exchange of mass, momentum and energy between forests and atmosphere, which in turn influence local and global carbon, water and other biogeochemical cycles.

Several experimental and numerical methods have identified canonical features of CSL turbulence. Poggi et al. (2004b)

conducted flume experiments over model canopies simulated by rigid rods and proposed a conceptual model for flow through uniform vegetated canopies which is shown in figure 1. The bottommost layer (region I) is dominated by von Kármán (VK) vortices produced in the wake space of the canopy. The mixing length in this zone is independent of canopy plant area index (PAI - a measure of foliage density of the canopy, defined as volume of foliage per volume of air) and only depends on the canopy diameter ($d_c$) and the vortex shedding frequency. Region II is the region close to the canopy top and characterized by

Kelvin-Helmholtz instabilities associated with the inflection point of the mean velocity profile. Momentum exchange in this region is governed by a superimposition of Kelvin-Helmholtz waves and boundary layer vortices. Incidentally this zone was described to be a juxtaposition of mixing layer and boundary layer vortical structures long ago by Raupach and Thom (1981) and Finnigan (2000). The characteristic length scale of this region depends on the strength of mean wind shear and the strength of the gradient of the vertical profile of the mean velocity. Region III, which is above the canopy, is governed by boundary

layer vortices the size and strength of which are proportional to the distance from the wall. These eddies 'see' the zero plane displacement surface as the 'wall' from which they grow. the zero plane displacement length is defined as the centroid of canopy momentum drag and shown as $d$ in figure 1.

Several other works have contributed to the understanding of this canonical structure of homogeneous canopy flows (Wilson and Shaw, 1977; Finnigan, 1979; Antonia, 1981; Raupach, 1981a; Denmead and Bradley, 1985; Baldocchi and Meyers, 1988;

Wilson, 1988; Gao et al., 1989; Holland, 1989; Shaw et al., 1989; Wilson, 1989; Shaw and Schumann, 1992; Kaimal and





Finnigan, 1994; Zhuang and Amiro, 1994; Raupach et al., 1996; Katul, 1998; Katul and Chang, 1999; Finnigan, 2000; Siqueira et al., 2000; Katul et al., 2004; Poggi et al., 2004b, a; Watanabe, 2004; Katul et al., 2006; Gavrilov et al., 2010, 2011; Chamecki, 2013; Katul et al., 2013; Dias-Junior et al., 2015), although recent focus in this field has shifted towards studying the effects of complex topography and heterogeneities such as edges and forest clearings. However, there are still gaps in the homogeneous

canopy turbulence literature which is the motivation for this study.

In numerical modeling of canopy turbulence, the canopy is represented (as a sink for momentum) by a drag force $F_c = C_d a U |U|$, where $C_d$ is the drag coefficient typically between $0.1 - 0.3$ (Banerjee et al., 2013), $a$ is the vertical leaf area density profile, the integration of which over the canopy height yields the aforementioned PAI; $U$ is the mean longitudinal velocity profile and $|U|$ is the mean wind speed profile. Both Reynolds averaged Navier Stokes (RANS) modeling (Katul et al., 2004;

Banerjee et al., 2013) and large eddy simulations (LES) of canopy turbulence (Shaw and Schumann, 1992) use this formulation. However, since the flow does not encounter real rigid structures as in the field or in lab experiments, the aforementioned von Kármán and Kelvin Helmholtz structures are not explicitly observed. Hence we want to answer the following questions:

– Do numerical simulations of canopy turbulence at least contain the signatures of these motions? Alternatively, is the drag formulation consistent with the phenomenological model as discussed earlier?

– If these signatures are obtained, do they strengthen if canopies and wake spaces are explicitly contained in simulations?

– How do these signatures vary with spatial variations of the canopy structure?

We use a number of techniques to address these questions, namely visualizations of turbulent fields and statistics, proper orthogonal decomposition, Shannon entropy, mutual information content (MIC) and synchronization analysis.

## 2   Theory

### 2.1   von Kármán motions in canopy turbulence

von Kármán vortices are associated with wake production at the horizontal plane close to the ground at the bottom most layer of the sub-canopy space. As the flow encounters a bluff body or a cylinder, succession of counter-rotating eddies break out from either side of the cylinder. In lower Reynolds numbers, these eddies are organized symmetrically in a laterally staggered pattern in the horizontal plane. As the Reynolds number increases, their spatial coherency is distorted. However, the Strouhal

number defined as $St = f_v d_c / |U|$, (where $f_v$ is the vortex shedding frequency) remains fairly uniform, between $0.18 - 0.21$ across a wide range of Reynolds number between 360 and $2 \times 10^5$ (Fey et al., 1998; Cava and Katul, 2008). Experimental evidence of von Kármán vortices in the canopy trunk space were provided by Seginer et al. (1976), Poggi et al. (2004b) and Poggi and Katul (2006). The von Kármán vortices were visualized by Poggi et al. (2004b) and Poggi and Katul (2006) at the height $z/h_c = 0.2$ (where $z$ is vertical height from the bottom and $h_c$ is the canopy height) using laser induced fluorescence

(LIF) experiments. With increasing Reynolds number, the vortices were distorted by turbulence, disturbance from sweep and ejection motions as well as obstruction from other canopy elements. However, they also observed the fairly stable Strouhal



number $St = 0.21$ which was linked to an injection of energy and a consequent kink at the velocity spectra near the vortex shedding frequency by a spectral short circuiting method (Cava and Katul, 2008).

Unfortunately this spectral signature cannot be observed in numerical simulations because of smoothening due to a larger grid resolution. The scale of processes associated with the production of the von Kármán vortices would be subgrid scale

process in a large eddy simulation if rigid trunk elements were present. The representation of the canopy as a drag effect allows the flow to enter the canopy (in reality the flow would go around the trunk), which further precludes the explicit observation of VK vortices in an LES.

Hence instead, we would like to look for increased activity and synchronous behavior of longitudinal ($u$) and cross stream ($v$) velocity series at the canopy wake space, since the VK motions are effectively organized motions of $u$ and $v$ velocities in

the $x - y$ plane close to the canopy bottom. In the data analysis section, we will use this weak and more generic definition to identify VK signatures.

## 2.2   Kelvin Helmholtz motions for canopy turbulence

Kelvin Helmholtz (KH) structures are characteristic of the mixing layer described by the inflected mean velocity profile close to the canopy top. This inflected profile is inviscidly unstable to small perturbations (Michalke, 1965; Raupach et al., 1996;

Drazin and Reid, 2004; Finnigan, 2000). The perturbations are often provided by sweeping motions originated in the surface layer above and the KH instability grows in the early development stages of the mixing layer. The KH structures evolve into connected roller type structures linked by braided regions (Rogers and Moser, 1992; Finnigan, 2000). The length-scale of the KH instability, dependent upon canopy drag and the inflection strength of the velocity profile is preserved in the fully developed turbulence. In large eddy simulations, it is hard to observe this development stage again because of subgrid scale averaging

and because of the fact that the data in the simulation spin up period is usually rejected and we only look at fully developed turbulence. Hence to look for remaining signatures of KH structures, we will look at increased activity and coupled dynamics of the longitudinal $u$ velocity and the vertical $w$ velocity. In this way, we construct a 'weak' or loose definition of the KH motions as organized motions of $u$ and $w$ in the $x - z$ plane close to the canopy top.

## 3   Large eddy simulations (LES)

Three sets of large eddy simulations are conducted - one with a horizontal homogeneous canopy, one with patched canopies and another with patched canopies with different spacing. The patched configurations allow us to investigate whether simulating canopies with the drag formulation permits the KH and VK motions to develop better if wake spaces are present. Figure 2 shows a mid-section of the simulated domain for two different patch configurations. The canopy height $h_c$ is 35 m. The domain size in all cases are $314m \times 314m \times 314m$. The grid resolution $\Delta$ is $0.98m \times 0.98m \times 0.98m$. The wind speed from the left

side is $7\mathrm{ms}^{-1}$. The plant area index (PAI) is $5\mathrm{m}^3\mathrm{m}^{-3}$ , characteristic of dense canopies and the plant area density profile is shown in the inset of figure 2. In both patched configurations, the canopy patches are of the same size : $5\Delta$ on both $x$ and $y$ directions. The same pattern for the canopy patches and the spaces between them are repeated for the entire domain which is



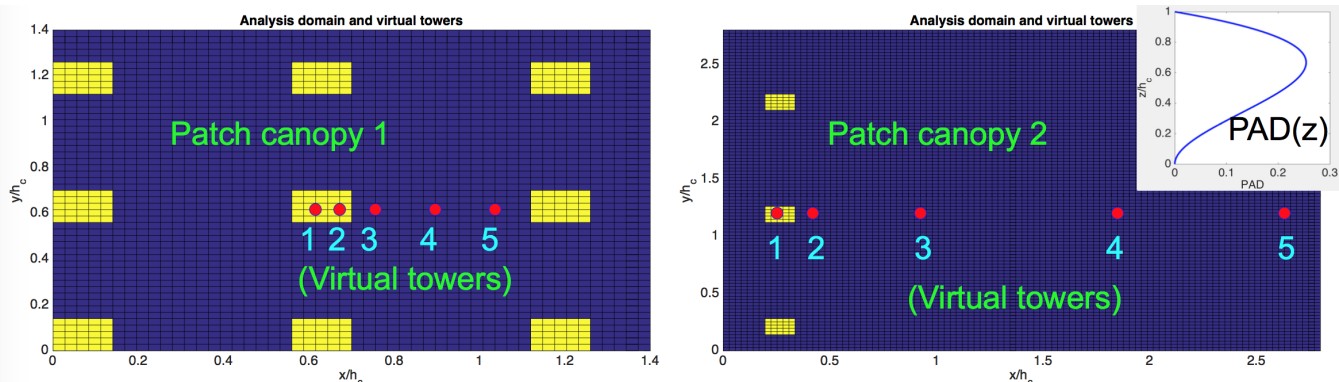

**Figure 2.** Patched canopy configurations simulated in the LES. Locations of virtual towers are shown as well. The inset shows the plant area density ($m^2 m^{-3}$) profile, the integration of which yields the PAI.

about the size $9h_c$ in each direction. The difference between the two sections shown in figure 2 are in terms of the spacing between the patches.

The PALM large eddy simulation (LES) model (Raasch and Schröter, 2001; Maronga et al., 2015) is used to simulate flow through the different canopy configurations. The representation of the canopy in the LES follows the standard distributed drag

parameterization (Shaw and Schumann, 1992; Watanabe, 2004; Patton et al., 2016) by adding an additional term in the momentum budget equations as $F_{di} = -C_d a |\overline{u}| \overline{u_i}$ where $a$ is a one sided frontal plant area density (PAD), $C_d$ is a dimensionless drag coefficient assumed to be 0.3 (Katul et al., 2004; Banerjee et al., 2013), $|\overline{u}|$ is the resolved wind speed and $\overline{u_i}$ is the resolved velocity component ($i = 1, 2, 3$, i.e. $u$, $v$ and $w$), i.e. the overbar denotes here implicit averaging over a grid box in LES. The effect of the canopy on the subgrid scale (SGS) turbulence is accounted for by adding a sink term to the prognostic

equation for the SGS turbulent kinetic energy ($e$) as $F_\epsilon = -2C_d a |\overline{u}| e$. For closure of the SGS covariance terms, PALM uses a 1.5 order closure developed by Deardorff (1980) and modified by Moeng and Wyngaard (1988) and Saiki et al. (2000), which assumes a gradient diffusion parameterization. The diffusivities associated with this gradient diffusion are parameterized using the subgrid scale turbulent kinetic energy (SGS-TKE) and includes a prognostic equation for the SGS-TKE. This SGS-TKE scheme after Deardorff (1980) is deemed to be an improvement over the more traditional Smagorinsky (1963) parameterization

since the SGS-TKE allows for a much better estimation for the velocity scale corresponding to the subgrid scale fluctuations (Maronga et al., 2015). Further details of the LES model can be found in the literature and are not discussed here (Shaw and Schumann, 1992; Watanabe, 2004; Maronga et al., 2015; Patton et al., 2016). Each simulation is run for 8200 s with a time step of 0.1 s, while the output of first 6400 s are discarded to achieve computational equilibrium. The last 30 minutes (1800 s) are used for data analysis here. 5 virtual towers are established in the simulation for the patch canopy cases as shown in figure

2, where data are saved at every vertical grid point at 20 Hz for 30 minutes. One such virtual tower is also established at the middle of the domain for the horizontal homogeneous case.





The simulation is driven by a geostrophic wind, which leads to a pressure gradient within the domain, and by the surface buoyancy flux, with a sensible and latent heat component. The boundary conditions at the bottom are Neumann for potential temperature and humidity, but Dirichlet for the velocity (no-slip). The lateral boundary conditions are periodic.

In the next subsections we discuss some of the techniques used in identifying the signatures of KH ($u$, $w$ motions close to canopy top) and VK ($u$, $v$ motions close to canopy bottom) motions, namely proper orthogonal decomposition (POD), Shannon entropy, mutual information content and synchronization analysis.

## 4   Data analysis

### 4.1   Proper orthogonal decomposition (POD)

Proper orthogonal decomposition is a technique that reveals the most energetic coherent structures in a flow field (Lumley, 1967; Berkooz et al., 1993; Pope, 2001; Graftieaux et al., 2001; Smith et al., 2005; Caver and Meyer, 2012; Calaf et al., 2013; Chen et al., 2013; Tirunagari, 2015; Banerjee et al., 2017). Multiple techniques can be found in the literature - however the method of snapshots is adopted in the current work which is one of the most simplistic. In this method, snapshots of the flow fields in time are used and they are decomposed with a new set of orthonormal basis functions so that they span the data space in the most optimal way. Mathematically this translates into finding the eigenvectors of the flow variables, which are called POD modes. The top few POD modes contain the majority of the energy of the system, revealing the most energetic coherent structures of the flow. Operationally, if two dimensional cuts of the flow field are used, such as a plane of $u(i,j)$ and $w(i,j)$ velocity fields, where $i$ and $j$ indicates the index of the grid points in $x$ and $z$ direction, let $V_k$ denotes one such ordered pair of $(u,w)_{i,j}^{(k)}$, where $k$ is the index of the snapshot in time. The POD analysis uses the following decomposition:

$$V^{(k)} = \sum_{m=1}^{M} c_m^{(k)} \phi_m. \tag{1}$$

where $\phi_m(i,j)$ are the orthonormal basis functions or the POD modes and $c_m^{(k)}$ are the eigenvalues or coefficients. One has to maximize the total kinetic energy of the system under the additional constraint that the POD modes need to be orthonormal (Banerjee et al., 2017). We use POD analysis using cuts of $u$, $v$ planes to look for signatures of VK motions close to the bottom and cuts of $u$, $w$ planes close to the canopy top to look for KH motion signatures.

### 4.2   Shannon entropy and mutual information content (MIC)

Shannon entropy is a topological measure from information theory that can be used as a tool to study flow organization and coherence. Wesson et al. (2003) used Shannon entropy to study the degree of flow organization in the canopy sub layer (CSL) and differentiate the dynamics of flow organization between the CSL and the atmospheric surface layer (ASL). Banerjee et al. (2017) used similar measures to study wave - turbulence - vegetation interaction under wind shear over a static water body.



Shannon entropy can be defined as

$$I_S = -\sum_i p_i \ln p_i \qquad (2)$$

while $p_i (i = 1, 2, ..., M)$ is a non-negative and normalized discrete probability distribution with additive properties, i.e.,

$$p_i \geq 0; \quad \sum p_i = 1; \quad p_{i \cup j \cup ...} = p_i + p_j + ... \qquad (3)$$

(Shannon, 1948; Wijesekera and Dillon, 1997; Wesson et al., 2003).

For the Shannon entropy to reflect the signature of the spectrum of a particular process, the spectrum can be defined as a sequence $\Psi_i (i = 1, ..., M)$ of $M$ numbers such that $M > 0$ and $p_i = \Psi_i (\sum_i \Psi_i)^{-1}$ (Wijesekera and Dillon, 1997). With this framework, the aforementioned probability distribution represents the particular process.

A white noise spectrum is devoid of any nonlinear organization and has the highest measure of Shannon entropy,

$$\Psi_i = 1; \quad p_i = M^{-1}; \quad I_{Sf} = \ln(M), \qquad (4)$$

so $0 < S_N = I_S / I_{Sf} < 1$ can be taken as a normalized Shannon entropy measure. This normalization also eliminates uncertainties between different experimental conditions under which the processes are recorded (Wesson et al., 2003). The physical significance of this non dimensional measure $S_N$ is that processes with a higher degree of complexity will exhibit a lower $S_N$. In terms of canopy turbulence, the CSL induces complex flow organization by imposing drag on the flow and thus CSL flows

are characterized by a lower degree of $S_N$ than flows in the ASL. In the current context, a lower $S_N$ on a particular velocity component will indicate higher flow organization. Thus lower $S_N$ of $u$ and $v$ motions close to the canopy bottom will carry signatures for VK motions and lower $S_N$ for $u$ and $w$ close to the canopy top will indicate KH signatures.

### 4.3 Mutual information content (MIC)

MIC is a another measure from information theory which can be used to study interdependence between two random variables.

The MIC between two random variables measures the information contained by one variable about the other:

$$I(s,q) = \sum_{i,j} p_{ij}(s,q) \ln p_{ij}(s,q) - \sum_i p_i(s) \ln p_i(s) - \sum_j p_j(q) \ln p_j(q), \qquad (5)$$

where $\sum_{i,j} p_{ij}(s,q) \ln p_{ij}(s,q)$ indicates the joint Shannon entropy between $s$ and $q$; $\sum_i p_i(s) \ln p_i(s)$ and $\sum_j p_j(q) \ln p_j(q)$ indicate Shannon entropy for s and q respectively (Banerjee et al., 2017).

Poggi et al. (2004c) used MIC to study the nonlinearity of the interactions between large and small scales in the canopy sub

layer. Because of wake formation between canopy elements, turbulent eddies in the CSL undergo several nonlinear processes such as vortex stretching and this allows for direct interactions between large and small scales associated with the spectral short-circuiting. This precludes the formation of direct energy cascade between large and small scales following a slope proportional to the turbulent kinetic energy (TKE) dissipation rate and instead TKE is directly transferred between large and small scales (seen as a deviation of power law TKE spectra inside the CSL). Poggi et al. (2004c) demonstrated that MIC can be used





as a direct measure of such nonlinear interactions and the CSL exhibits a much higher interaction between large and small scales inside the canopy than above it. Now Poggi et al. (2004c) used flume experiments with rigid rods to study MIC and had the advantage of having well defined length scales (such as rod diameter or spacing) to differentiate between large and small scales. This advantage of having a well defined length scale is absent while using the drag formulation in numerical models.

So we use an energy filtering criterion based on a Lorentz curve analysis as presented in Banerjee et al. (2015). The energy filtering criterion follows the same philosophy employed in POD analysis, that most of the energy of a physical process is contained by the top few wave numbers (the velocity time series at every grid point are Fourier transformed to find the energy associated with every wavenumber. These wavenumbers are sorted in the descending order of energy containment, and top most energy containing modes are filtered out and transformed back to the real space. These highest energy containing modes

can be associated with the largest length scales of the process). To look for signatures of KH and VK motions, we will look for MIC between large and small scales for individual velocity components. Higher degree of MIC should be interpreted as flow organization imposed by canopy elements on individual velocity components, which is the essential physics behind KH and VK motions, as discussed earlier.

### 4.4   Synchronization analysis

For the analysis of the coupling between two dynamical variables, we use a synchronization analysis. To be able to do this, we additionally interpret the KH and VK motions as the dynamics between two independent oscillators. Thus KH motions can be modeled as the coupled phase dynamics between $u$ and $w$ components and VK motions can be modeled as the coupled phase dynamics between $u$ and $v$ components. To this end, the protophase of the original signal is first transformed into a (reconstructed) phase. As advocated by Kralemann et al. (2008), this transformation makes the comparison of the signals more

robust. The DAMOCO software (Rosenblum and Pikovsky, 2001; Kralemann et al., 2007, 2008) is a MATLAB package that performs this transformation and allows to compute a synchronization index ($Syn$) between weakly to moderately coupled self-sustained oscillators whose dynamics obey

$$\dot{\phi}_1 = \omega_1 + q^{(1)}(\phi_1, \phi_2), \tag{6}$$
$$\dot{\phi}_2 = \omega_2 + q^{(2)}(\phi_2, \phi_1). \tag{7}$$

Here $\phi_i$ are the phases of the oscillators, $\omega_i$ are the natural frequency and $q^{(i)}$ are the coupling functions. The synchronization index is a measure for the phase coupling of the dynamic variables, where a constant phase difference (phase–locking) would yield $Syn = 1$ and uncoupled oscillators have $Syn = 0$ (Kralemann et al. 2008). For our purpose, the synchronization index can express if the two signals are indeed coupled or if they oscillate independently. Hence for KH motions, we expect a higher $Syn$ between $u$ and $w$ motions close to the canopy top. For VK motions, there should be a higher $Syn$ between $u$ and $v$ close

to the bottom. Note that this analysis only considers the phase dynamics of the oscillators (velocity series) and ignores their amplitudes, which makes this analysis different from covariance calculations.



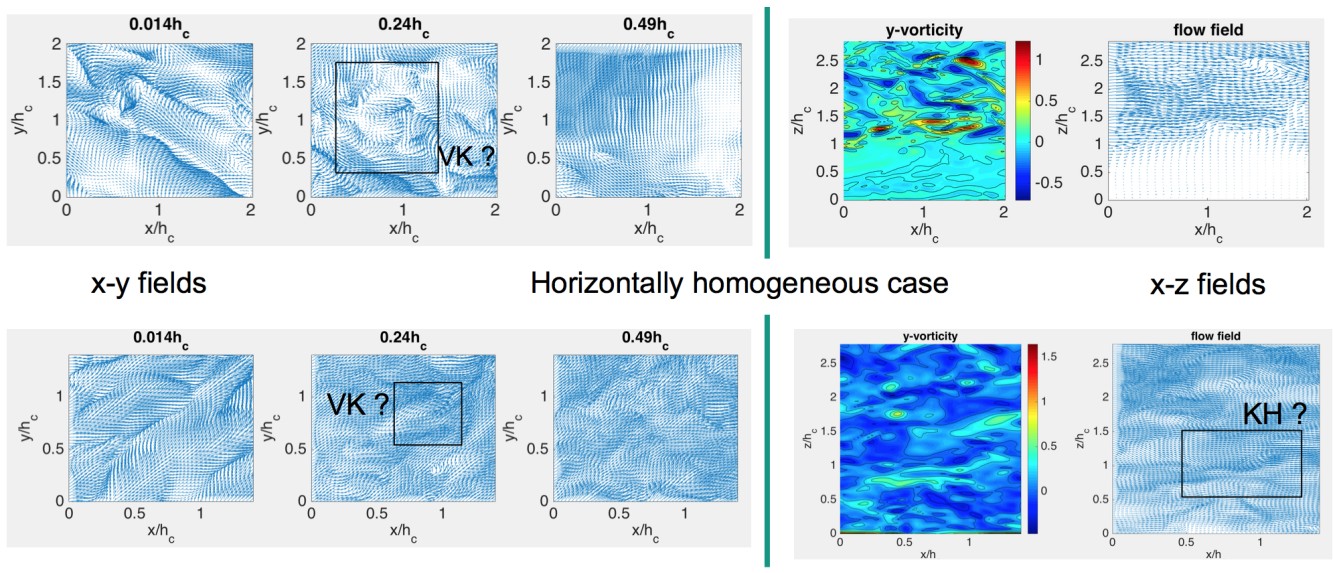

**Figure 3.** LES outputs of instantaneous flow fields for the horizontally homogeneous case and the path canopy configuration 1. Left panels (left of the solid line) show x-y velocity fields and right panels show x-z planes.

## 5    Results and discussions

### 5.1    LES outputs

Figure 3 shows instantaneous LES outputs of flow fields for the horizontally homogeneous case (top row) and the patch canopy configuration 1 (bottom row). The left panels show x-y quiver fields for both cases at three different heights : $0.014h_c$, $0.24h_c$ and $0.49h_c$. The quiver plots show several vortical structures as enclosed in boxes, however, it cannot be confidently stated that these are signatures of VK motions. The panels on the right side show x-z planes for both the horizontally homogeneous and patch canopy 1 cases, plotting the y component of vorticity and $u - w$ quiver plots. Again, these fields show weak signatures of billow type structures (marked by box), but it cannot be confidently stated that they are signatures of KH motions. However, these LES field outputs conform to other LES results in the literature (Patton et al., 2016). Close to the canopy bottom, the coherent structures go through vorticity stretching - so elongated streaks are visible in the x-y fields at $0.014h_c$. The streaks give way to more roll type structures close to $0.25h_c$ and they are destroyed as one moves to the canopy top. A richer dynamics of x-y vortical fields are visible for the patch configurations because of the patch and wake spaces. In terms of y-vorticity in the x-z plane, strong and alternate positive and negative structures are visible close to the canopy top where KH motions are expected. Overall, these LES outputs are only provide qualitative weak detection of KH-VK motions.

Figure 4 shows time averaged turbulent statistics computed at the virtual towers as described in section 3. The leftmost column shows vertical profiles of quantities for the horizontally homogeneous (HH) case, the second column for the patch





**Figure 4.** Time averaged and normalized vertical profiles of turbulent statistics computed over the virtual towers for the horizontally homogeneous (HH) case, patch canopy 1 and patch canopy configuration 2. Blue, cyan, green, yellow and red indicate the 5 virtual towers in their order (T1 to T5) respectively.





canopy configuration 1 and the third column for the patch canopy configuration 2. The x axis label indicates the plotted quantity, all of which are normalized by their corresponding value at the canopy top. For the HH case, there is only one plot because there is only one virtual tower. For the patch canopy cases, the colors blue, cyan, green, yellow and red indicate the 5 virtual towers in their order (T1 to T5) respectively. Note that for patch canopy 1, towers 1 and 2 are inside the canopy and 3,4,5 are in the wake space with increasing distance from the canopy. For patch canopy configuration 2, tower 1 is inside the canopy and towers 2,3,4,5 are in the wake space. Tower 5 is close the next canopy patch (refer to figure 2). There are several interesting features in the turbulent statistics that conform to existing knowledge in the canopy turbulence literature:

– The first two rows show mean profiles for $V$ and $W$ velocity. For the HH case, we see the attenuated mean $V$ velocity profiles inside the canopy and the classic log profile above the canopy. For the patch cases, tower 1 and 2 show the attenuated profile inside the canopy and they reach their free surface layer log profile in the canopy space. For the mean vertical velocity profile $W$, the profiles inside and outside the canopy show significant difference possibly due to edge effects (Banerjee et al., 2013). One interesting observation is increased $W$ magnitude close to the canopy top for the HH case, patch 1 and the tower 5 for patch 2, which is close to the next canopy, which are relevant for signatures of KH motion. However, some of the signatures of KH and VK motions can be destroyed because of time averaging over alternate positive and negative vortical structures.

– The third and fourth rows show normalized profiles for the velocity variances $\sigma_v$ and $\sigma_w$ respectively. Profiles for the patch canopy cases show an increased $\sigma_v$ close to the canopy bottom which could be a signature of VK activity. The $\sigma_v$ profiles on towers far away from the canopy edges in patch 2 configuration do not show this increased activity in $\sigma_v$ close to the canopy bottom. For all cases inside the canopy, the $\sigma_w$ profiles show an attenuated profile. Far from the edges, the attenuation is not visible.

– The fifth row (last on the left side) and the sixth row (first on the right side) show mean Reynolds stress profiles, in terms of $\overline{u'w'}$ (indicative of coupled $u$ and $w$ activity, thus signature for KH motions) and $\overline{u'v'}$ (indicative of coupled $u$ and $v$ activity, thus signature for VK motions). For all cases, close to the canopy top, $\overline{u'w'}$ is high, indicating of possible KH motions, although their variations change with respect to distance to the canopy edge. The $\overline{u'v'}$ profiles do not show such clear variations. Although the tower inside the canopy for patch 1 and the ones inside and close to the canopy for patch 2 show local maxima, which can be interpreted as signatures of VK activity.

– The last two rows on the right side show profiles of integral length scales for the $v$ and $w$ velocities. They are calculated by integrating the autocorrelation functions up to the first zero-crossing, which is a standard procedure and not repeated here. The integral scales indicate the most energetic eddies. Local maxima of $I_v$ are only observed for the patch 2 configurations. However, profiles for the patch canopy 1 configurations show local maxima for $I_w$, which can be interpreted as a signature of KH motion.





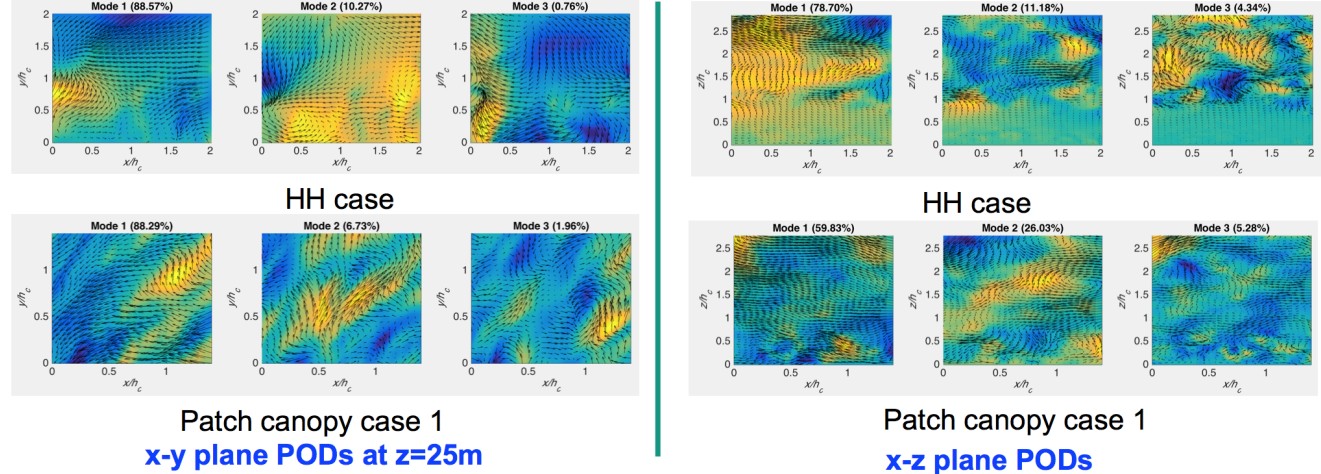

**Figure 5.** Top three POD modes for the HH case (top) and patch canopy case 1 (bottom). Left panels show an x-y plane at $z = 35$m and right panels show x-z planes. x-y planes are color coded by $\phi_y$ components and x-z planes are color coded by $\phi_z$ components.

## 5.2 POD outputs

Figure 5 shows top three POD modes along with their energy content for the HH case (top) and the patch canopy case 1 (bottom). The left panels (left of the vertical bar) show POD modes for an x-y plane at $z = 35$m, color coded by the $v$ component of the POD modes. Right panels show an x-z plane (color coded by the $w$ component of the POD modes) at the same section where the virtual towers are located. The POD modes can help us to visualize the top most energetic structures at these particular planes and if KH and VK motions are present, they should be revealed in a much clearer way than what can be detected by visualizing instantaneous vorticity fields. Looking at the left panel, top two POD modes contain almost all of the energy of the flow field and both POD modes show alternate positive and negative vortical structures, which are clear signatures of VK motions. For the patch cases, several of these alternate positive and negative vortical structures are present, also indicating of strong VK signatures. Looking at the right panels, the POD modes reveal strong alternate positive and negative billows close to the canopy top, which is a strong signature of KH motions. These signatures are visible more strongly for the HH configuration compared to the patch configuration. The second largest POD modes associated with KH motions contain significantly more energy compared to the second POD mode for VK motions, highlighting the difference between the physical mechanisms behind these two processes.

## 5.3 Shannon entropy outputs

Figure 6 shows non dimensional Shannon entropy profiles for the HH case (left column), patch case 1 (middle column) and patch case 2 (right column). The top, middle and bottom rows show Shannon entropy for $u$, $v$ and $w$ velocity components at the virtual towers. As observed, the profiles are quite noisy and they change depending upon the location of the virtual tower. It





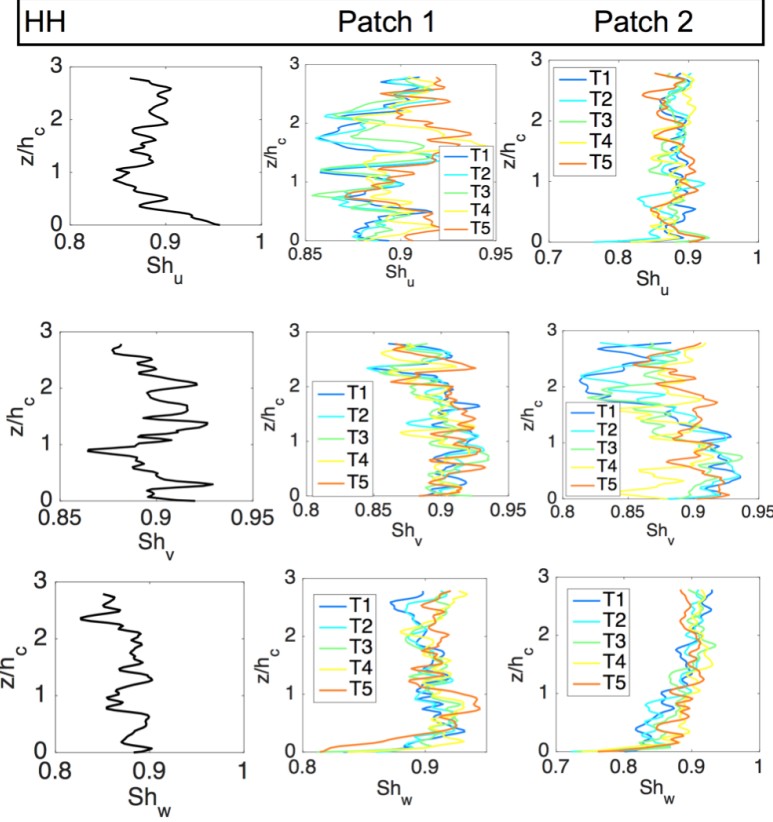

**Figure 6.** Vertical profiles of Shannon entropy for the HH case (left column), patch case 1 (middle column) and patch case 2 (right column). The top, middle and bottom rows show Shannon entropy for $u$, $v$ and $w$ velocity components at the virtual towers.

is important to recall that a lower Shannon entropy indicates high degree of flow organization. $Sh_u$ profiles show local minima close to the canopy top which is consistent with KH motions. Similar local minima are also observed for locations inside the canopy for the patch configurations. The signatures of VK motions through local minima of $Sh_v$ are not very clearly visible in most of the profiles.

## 5.4 MIC profiles

Figure 7 shows MIC between large and small scales for for the HH case (left column), patch case 1 (middle column) and patch case 2 (right column). The top, middle and bottom rows show MIC for $u$, $v$ and $w$ velocity components at the virtual towers. There are a few interesting features that can be observed in figure 7. For the HH case, the MIC between large and small scales for all velocity components inside the canopy is much larger than above the canopy, which is consistent with the results published by Poggi et al. (2004c). Also $MIC_v$ has a local maxima close the the canopy bottom, a strong signature for VK motions. $MIC_w$ has a local maxima close to the canopy top, a strong signature for KH motion. The patch canopy case 1 with



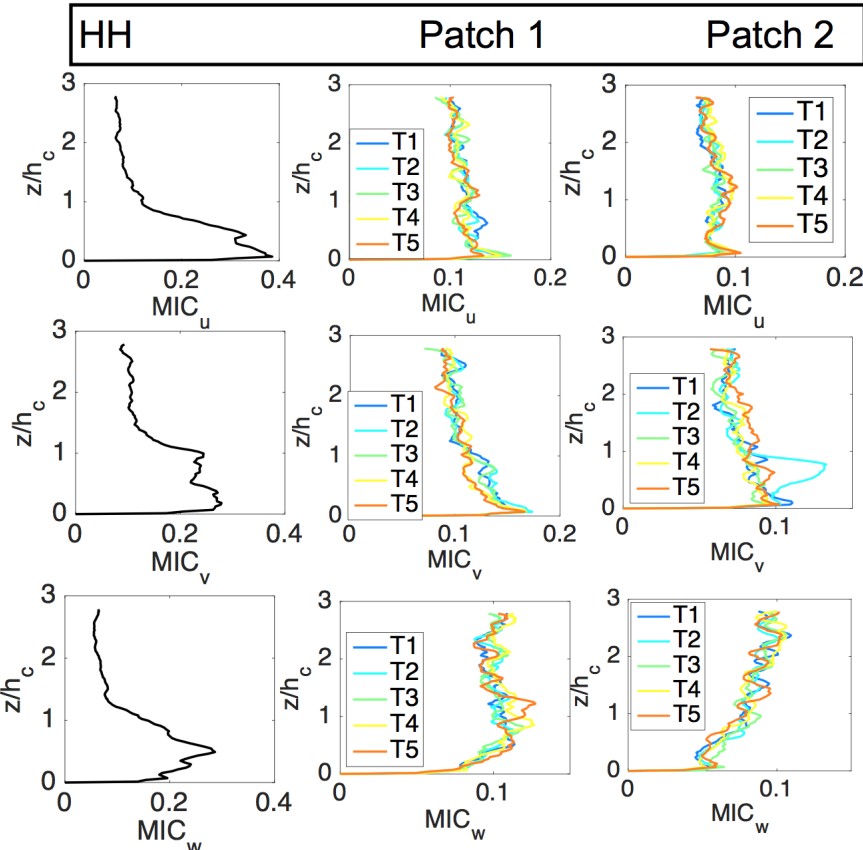

**Figure 7.** Vertical profiles of mutual information content (MIC) between large and small scales for the HH case (left column), patch case 1 (middle column) and patch case 2 (right column). The top, middle and bottom rows show MIC for $u$, $v$ and $w$ velocity components at the virtual towers.

smaller spacing between the patches follow similar patterns for $MIC_v$ like the HH case, but the signatures vary depending on the location of the virtual tower and they are weaker than the HH case. Interestingly, the $MIC_w$ profiles are quite different for the patch canopy configurations. Indicating that the patch and wake configurations have more impact on KH motions than VK motions.

## 5.5 Synchronization analysis results

Figure 8 show vertical profiles of the synchronization indices between $u$ and $v$ motions (top row) and $u$ and $w$ motions (bottom row). Left column is for the HH configuration, the middle and the right column are for the patch 1 and patch 2 configurations. Recall that higher synchronization index between two velocity components means organized synchronous motion between these two components. Thus high $Syn_{uv}$ close to the bottom should indicate VK motions and and high $Syn_{uw}$ motions close to the canopy top means stronger KH motions. As observed in figure 8, the $Syn$ profiles are quite noisy for the HH case.





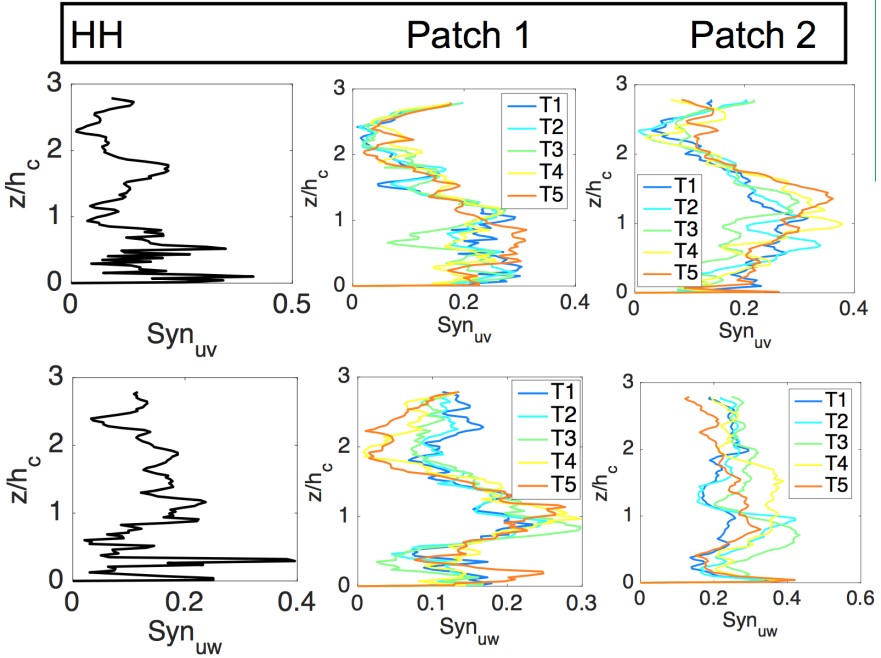

**Figure 8.** Vertical profiles of the synchronization indices between $u$ and $v$ motions (top row) and $u$ and $w$ motions (bottom row). Left column is for the HH configuration, the middle and the right column are for the patch 1 and patch 2 configurations.

But for the patch configuration 1, there are synchronous motions between $u$ and $v$ components close to the bottom, indicating of VK motions. Some locations close to the canopy path for the patch configuration 2 shows signatures of VK motion. The synchronization between $u$ and $w$ is quite strong for the patch 1 cases close to the canopy top, indicating strong KH motions. The patch 2 cases also show stronger KH motions close to the canopy top.

5 # 6 Conclusions

Conceptual models for turbulent flow inside vegetation canopies are based on lab and field experiments with real vegetation and obstacles. These experimental conditions create well defined von Kármán (VK) vortices in the horizontal plane associated with wake production and short circuiting of energy between large and small scales of motion. The inflected vertical velocity profile also gives rise to Kelvin Helmholtz (KH) instabilities in the vertical plane, which are billow type structures close to the canopy

10 top. Based on these motions, the vertical extent of the canopy sub layer can be divided into three different regimes: the bottom layer dominated by von Kármán wakes, a middle layer characterized by a superimposition of a mixed layer associated with Kelvin Helmholtz instabilities and boundary layer eddies and the top of the canopy transitioning into the atmospheric surface layer above. This three layer structure has been the backbone of the phenomenological model of canopy turbulence. However, numerical simulations of canopy turbulence represent the canopy with an extended drag formulation and the flow can intercept





the canopy. In the current manuscript, we want to reconcile this representation with the aforementioned phenomenological model and ask the following questions:

- Do numerical simulations of canopy turbulence at least contain the signatures of these KH and VK motions? Alternatively is the drag formulation consistent with the phenomenological model as discussed earlier?

- If these signatures are obtained, do these they strengthen if canopies and wake spaces are explicitly contained in simulations?

- How do these signatures vary with spatial variations of the canopy structure?

To answer these questions, we have conducted large eddy simulations over three different canopy configurations - a horizontally homogeneous canopy, a patch canopy configuration with spaces between the patches for wakes to develop better and
10    another patch canopy configuration with larger spaces between the patches. we positioned virtual towers at strategic locations, where data were recorded at high frequency (20 Hz) over a half hour period - similar to micrometeorological observations. We used visualizations of turbulent flow fields and turbulent statistics to gain insight into these structures and used a couple of non-traditional measures from information theory in order to reveal more details, namely Shannon entropy and mutual information content between large and small scale motions. Another method of revealing synchronization between coupled oscillators is
15    used to reveal more details about this motions. In order to interpret the results, we have proposed a 'weak' alternate definition for the KH and VK motions. KH motions are interpreted as coupled $u - w$ velocity components close to the canopy top; and VK motions are interpreted as coupled $u - v$ motions close to the canopy bottom. Using these unique methods, we can answer the aforementioned questions:

- We can conclude that the drag formulations contain the signatures of KH and VK motions at least in the weaker sense.
20       different techniques reveal the signatures of these motions in different degrees of magnitude. It is often the case that signatures of KH motion are captured more strongly than VK motions in general. Also techniques which measure the coupled dynamics of velocity components compared to their individual measures show stronger signature.

- Surprisingly, the horizontally homogeneous cases often contain a stronger signature of both motions. The patch canopy cases also contain these signatures weakly but edge effects often impose additional dynamics which can mask the KH
25       and VK motions. Hence it can be concluded that having extended canopy patches and wake spaces does not necessarily substitute having real obstacles and wakes in the flow field. Since the flow cannot go around the obstacles, there are strong edge effects that are not visible in experiments with real obstacles.

- The geometry of patch and wake configurations definitely alter the signature of KH and VK motions. In case of smaller spacing between patches, there are stronger edge effects on both sides, namely entry and exit from a canopy edge. In
30       case of large spacing, the flow equilibrates to a regular boundary layer flow before being subjected again to the effect of the next canopy edge. In both cases, signatures of KH and VK motions are masked due to these influences.



To summarize, numerical simulations of canopy turbulence with the extended drag formulation cannot capture the explicit strong signatures of KH and VK motions as captured in real experiments. However a horizontally homogeneous canopy with a distributed momentum drag contains somewhat weak signatures of both types of motions (stronger signatures of KH motions). A patch and wake type configuration even with high spatial resolution does not necessarily capture these motions better. These

5 results are deemed to be an important step to evaluate the current state of the art in the field and highlights intricate features of the complicated nature of CSL turbulence. Future work will attempt to place real obstacles in the form of solid stems to improve the reliability of numerical simulations of CSL turbulence.

*Acknowledgements.* This research was supported by the German Research Foundation (DFG) as part of the project "Climate feedbacks and benefits of semi-arid forests (CliFF)" and the project "Capturing all relevant scales of biosphere-atmosphere exchange - the enigmatic energy

10 balance closure problem", which is funded by the Helmholtz-Association through the President's Initiative and Networking Fund, and by KIT.





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
