# Peer review of "Revisiting Kelvin Helmholtz Instabilities and von Kármán Vortices in Canopy Turbulence."

_Hydrology and Earth System Sciences, 2017_

## Referee Comment (RC1) · Anonymous Referee #1 · 24 Nov 2017

General comments: The paper presents results from LES of canopy patches with 2 different configurations of spacing between canopy patches. The canopy patches are modelled with a drag formulation in the LES. The stated goal is to investigate if von Karman streets within the canopy, and Kelvin-Helmholtz instabilities above the canopy, are obtained in the LES despite the drag formulation. Several rather innovative analysis methods (POD, Shannon entropy and mutual information criteria, synchronization analysis) are used as a mean to investigate the presence of von Karman streets within the canopy and Kelvin-Helmholtz instabilities above the canopy. The conclusions remain however very hypothetical and too quickly stated. Overall, the paper lacks rigour in its presentation (description of the simulations as well as the methods) and in the interpretation of results. The analysis methods are (at least for some) not standard in

the community and could be introduced more clearly.

Specific comments: - LES description: The setting description of the canopy patches is confusing and no information is given on the spacing between patches for the two different settings, nor on non-dimensional distances of virtual towers to the vegetation patches. Yet the effect of spacing between patches is stated as a major question to be addressed by the analysis. This question should be addressed rigorously with non-dimensional results with a range of spacing and normalized distance to a patch. The equation set could be written fully. The simulations include scalars (temperature and water vapour) but no mention is made on stability conditions, which will have a strong effect on structures. As described, the simulations would not be reproducible.

- The alternative, so-called 'weak' definition of the VK and KH motions is not introduced properly and should be presented with rigorous hypotheses.

- Method descriptions: The Lorentz curve method is not explained. A parenthesis tries to summarize the idea in a rushed way that is not helpful nor reader friendly. The synchronization analysis is poorly justified in the sense that using a model based on two oscillators to identify the presence of KH or VK phenomena in the simulation outputs is not supported by appropriate hypothesis. Do the authors assume linear wave theory to model the KH or VK phenomena using two oscillators? What is the protophase?

- Interpretation of results: overall, the interpretation of results lacks rigour. The values of the different indicators (Shannon entropy, $$, local maxima of $I_w$ ) are not sufficiently linked to physical processes and are concluded to be signs of KH and VK activity. This is not convincing and comes out as overstatements.

- POD analysis: Patch 2 scenario is suddenly abandoned in the POD results section without justification. That the POD results are conclusive for the presence of von Karman street is not so clear from the figures.

- MIC: large and small scales are poorly defined.

- Virtual towers of 20Hz frequency are used to compare with field results, although field results are never discussed. Is there a reason why the authors decided to use a 30 minute period rather than stationary, less noisy results?

―――――――――――――――

---

## Referee Comment (RC2) · Anonymous Referee #2 · 27 Nov 2017

**General comments:**

In this work, it is investigated the presence of von Kármán (VK) vortices and Kelvin-Helmholtz (KH) instabilities in Large-Eddy Simulation (LES) when the common distributed drag force approach is used to represent plant canopies. Three different simulations were performed: one with horizontally homogeneous drag force, and two with a different set of patched areas with drag force. Four different types of data analyses were used to characterize the presence of coherent structures in the LES flow field, namely proper orthogonal decomposition, Shannon entropy, mutual information content and synchronization analysis. In these analyses, several features were pointed out as indicators of the presence of coherent structures. It is concluded that a weak

signature of these coherent structures is present in the drag force formulation of plant canopies.

Overall, the motivation of the work is interesting (how realistic the drag force formulation of canopy in LES is), but this work presented a superficial and unconvincing evaluation of the signature of coherent structures in the simulation. Also, it is not provided any comparison of the LES results with data from a real canopy, which in my view is highly needed. I would like to point out some specific issues and questions that could help to improve the work.

**Specific comments:**

1. Abstract: it is already clear from the abstract that the work has more motivation than actual results or conclusions. After improving the content of the manuscript, the abstract should focus more on the results and conclusions of the work.

2. Introduction: a good description of the canopy layers, coherent structures and drag formulation is given, but the review on the current state of the art is poor. Paragraph between the lines 18 of page 2 and 5 of page 3 is a clear example. How do all of these works have contributed to the understanding of the canonical structure of homogeneous canopy? What did they find, and what didn't they find? Which gaps are these? More specifically, what kind of information regarding the signature of coherent structures in the canopy did they find, that could help to interpret your results? Is this the very first work testing how realist the drag force formulation is? Given how many times it has been used, I highly doubt that, but if this is the case, you should state it clearly.

3. Methodology: I am not convinced by this "weak and more generic" or "weak and loose" definitions of VK and KH motions. What are the basis for them? Have they

been used as surrogate for coherent structures before? I believe there will be an enhanced motion in these directions due only to the difference in the flow caused by the drag, even if there is no coherent structure (which should be captured by the drag formulation). I expect, for example, that this weak signature may represent purely local fluxes that has little to do with coherent structures. Is there a way to differentiate purely local flux to the true coherent structure signature? I believe this is a difficult task, but if you are claiming that there is some weak signature of coherent structures in the drag formulation, I believe you should be able to show this better.

4. Results: I'm very confused about how it is possible to have VK vortices in the horizontally homogeneous case. What is the mechanism of creation of them, if there is no discontinuity in the horizontal plane? How do you know these are not just "regular" turbulent motion (not created by coherent structures), but instead these are coherent structures? I would expect no VK vortices in the horizontally homogeneous case, and *maybe* something similar to VK vortices in the patches case due to the discontinuity (even though I believe in reality they are created by the complete blocking of the flow by plant trunks, and may not form from the reduced flow by a patch with drag). The KH vortices, yes, they are created by the inflection point which is present in the drag formulation, so they can be present in all cases (although in different ways based on the intensity of the inflection point). In order to differentiate "regular" turbulent motion from coherent structures, maybe a simulation without canopy could be used. Those snap-shots of Figure 3 do not show me anything. I cannot even see what is different inside the squares with the "VK?" from the rest of the plot. And if it is not conclusive, why show it?

5. Overall, in order to argue that any feature is due to the canopy presence, it is good to compare it with the case without canopy. Also, it is highly needed a direct comparison of LES results with cases in the literature where the coherent structures were really present. For example, you should compare LES statistics

with results from wind tunnel or field data where a real canopy was present, and point out which features are present in the LES and which are not. Only after that, you could show these new analyses that (I believe) nobody has done before, and show how they improve the interpretation of your results. So far I have no way of telling what are you actually showing with these new analyses, and if the features that you found are actually any weak signature of coherent structures or something else. I think that you reached the conclusion that those features are from coherent structures too fast and without enough evidence.

6. Conclusions: if there are significant differences between the drag model and the real case, it should be discussed what are the implications of these differences for all the studies that are done with LES.

---

## Referee Comment (RC3) · Anonymous Referee #3 · 29 Nov 2017

General Comments:

The paper evaluates the performance of a canopy model using a drag formula in the large eddy simulation (LES). Particularly, the existence of two key features –Kelvin Helmholtz (KH) waves and von Karman (VK) vortices – over the three canopy configurations are examined in this work. Several innovative approaches (proper orthogonal decomposition, Shannon entropy, Mutual information and synchronization analysis) are introduced to analyze the LES data. The conclusion confirms the existence of both phenomena in canopy flow simulations. However, overall, the analyses based on new approaches are incomplete and lack physical interpretation.

Specific Comments:

Major (The authors need to well address these comments before the paper can be

published):

1. The paper lacks some important information about the description of analysis approach. These include the lack of (i) the physical interpretation of mutual information content (MIC), (ii) the mathematical equation and physical meaning of synchronization analysis, and (iii) the approach to split the large scale and small scale in MIC analysis.

2. The description of LES setup is incomplete. For example, the paper doesn't provide some key parameters of two patched configurations (e.g. the separation between patches) and the reason to select those key parameters. The reasons to choose different locations of virtual towers between two patched configurations are not given – Why the number of virtual towers inside selected canopy patches are different between these two configurations? The detailed description of initial conditions is missing.

3. The use of virtual tower to mimic observations in lab or field experiment is interesting, but weakened by the lack of comparison with data from observations. This makes the conclusions much weaker, especially due to the use of innovative analysis approaches without any validation. The use of those approaches need to answer the following questions before being applied in a such case: can KH waves and VK vortices be interpreted using these approaches in a real case? and if they can, whether these phenomena can be observed in LES results?

4. The KH and VK phenomena can be identified from many traditional approaches including quadrant analysis, skewness analysis, variable-interval time averaging, wavelet transform, two-point space-time correlation tensor, etc. (a good review about these approaches is in Finnigan (2000)). However, these methods are not being used in current analysis. Why? For example, the following questions is critical: can KH and VK be identified using traditional approaches in current simulations? compared to these traditional approaches, what are the advantages of using these innovative methods? If there are no advantages, why bother?

5. The turbulent flow should be more or less symmetric with respect to the central plane of each canopy patch in streamwise direction. Thus, the turbulent flux  should be more or less zero at the virtual towers located on these planes. However, in Figure

4, the turbulent flux  is non-zero in all three cases. This phenomenon is not explained.

6. Some panels of figures are not analyzed in current manuscript and should not appear in the paper. These panels include the length scale of u in Figure 4 and third column of Figure 5.

Minor:

1. The resolution of Figure 3 and Figure 5 are too low in current manuscript which make them difficult to view. Meanwhile, the ranges of coordinates and color bar in both figures are not the same from case to case.

2. The title of section 4.3 is already included in title of section 4.2 and need to be corrected.

3. The index i shouldn't be used repetitively in equation 'pi=...' in page 7 line 7.

4. New variable V in page 11 line 8 need to be defined before being used.

5. Some other text typos and format problem associate with units need to be corrected.

Finnigan, J.: Turbulence in plant canopies, Annual review of fluid mechanics, 32.1, 519-571, 2000.

---

## Author Comment (AC1) · 13 Mar 2018

General comments:

**The paper presents results from LES of canopy patches with 2 different configurations of spacing between canopy patches. The canopy patches are modeled with a drag formulation in the LES. The stated goal is to investigate if von Karman streets within the canopy, and Kelvin-Helmholtz instabilities above the canopy, are obtained in the LES despite the drag formulation. Several rather innovative analysis methods (POD, Shannon entropy and mutual information criteria, synchronization analysis) are used as a mean to investigate the presence of von Karman streets within the canopy and Kelvin-Helmholtz instabilities above the canopy. The conclusions remain however very hypothetical and too quickly stated. Overall, the paper lacks rigor in its presentation (description of the simulations as well as the methods) and in the interpretation of results. The analysis methods are (at least for some) not standard in the community and could be introduced more clearly.**

We thank the reviewer for the constructive comments. We have improved the manuscript based on Your suggestions.

Specific comments:

**LES description: The setting description of the canopy patches is confusing and no information is given on the spacing between patches for the two different settings, nor on non-dimensional distances of virtual towers to the vegetation patches. Yet the effect of spacing between patches is stated as a major question to be addressed by the analysis.**

Thanks for pointing out. The spacing between the patches is indeed the important difference between the two configurations and they are now specified explicitly.

**This question should be addressed rigorously with non- dimensional results with a range of spacing and normalized distance to a patch.**

This is a great suggestion, however in a problem involving canopy patches (roughness transitions), there are a number of different length scales that can be invoked. So showing the results normalized by the canopy height - the only fixed length scale in the problem across different cases makes more sense. This is what we have done in figure 4.

**The equation set could be written fully. The simulations include scalars (temperature and water vapour) but no mention is made on stability conditions, which will have a strong effect on structures. As described, the simulations would not be reproducible.**

Thanks for this suggestion, we have created an appendix section where the full equations and boundary conditions are described in detail for reproducibility. There have been other significant works on the effect of stability and as Patton 2016 (referenced in the paper) notes, there is significant switching between cell and roll type structures as convective conditions change, which can be very well observed by visualizing w' fluctuations. However, in this work, variations of stability will add further variations on top of what are observed because of the three different configurations. Moreover, in this work we are more interested in simultaneous variations of different velocity components, so stability is held fixed. However the thermal and moisture boundary conditions are now mentioned in more detail.

- **The alternative, so-called 'weak' definition of the VK and KH motions is not introduced properly and should be presented with rigorous hypotheses.**

It is a widely accepted fact in atmospheric and engineering applications that velocity inflections are associated with KH type instabilities. From literature review it appeared to us that the presence of KHI is somewhat taken for granted in for canopy turbulence. For example, the well-known review paper Finnigan 2000 summarizes results from Raupach 1996 which calculated mean streamwise separation of coherent eddies near canopy top and presented a conceptual model which looks like figure 1.

[Figure]

Figure 1: Conceptual diagram of KHI as shown in Finnigan 2000.

They also conclude that although these structures evolve throughout the stages of turbulence development, the streamwise wavelengths are preserved. This motivates our current work in the sense that we want to confirm the presence of such structures directly from the velocity signals instead of the more abstract turbulence statistics that are usually found in the literature.
    Similarly for the VK structures, it is an accepted fact that they evolve in a horizontal place across the flow behind the obstacle and for canopy turbulence, this is taken for granted although it is only the lab experiments with real obstacles that observe these. We wanted to examine how realistic are these assumptions for the drag formulation. The so called weak definitions are thus not found in the literature, and we construct them in the paper. What we mean is we want a more direct evidence from the velocity records themselves. We describe this in more clearly and also remove the terms 'weak definition' to avoid confusion.

**Method descriptions: The Lorentz curve method is not explained. A parenthesis tries to summarize the idea in a rushed way that is not helpful nor reader friendly.**

Although we had referenced it from an older paper, we present in an appendix for the sake of completion.

**The synchronization analysis is poorly justified in the sense that using a model based on two oscillators to identify the presence of KH or VK phenomena in the simulation outputs is not supported by appropriate hypothesis. Do the authors assume linear wave theory to model the KH or VK phenomena using two oscillators? What is the protophase**?

We agree that this synchronization analysis is not found in the canopy turbulence literature. We first introduced in another paper (De Roo 2017, JAS) and we think this is a powerful technique to understand amplitude independent, phase only synchronization between weakly coupled oscillators. The velocity components are not completely independent as they are modulated by the same physical mechanism, although they are orthogonal to each other. This makes them perfect candidates as weakly coupled oscillators as described by Rosenblum 2001. We have described this in more detail in the text and created an appendix to describe the method in more detail, including the description the protophase.

**Interpretation of results: overall, the interpretation of results lacks rigor. The values of the different indicators (Shannon entropy, , local maxima of Iw ) are not sufficiently linked to physical processes and are concluded to be signs of KH and VK activity. This is not convincing and comes out as overstatements.**

Thanks for pointing out. We agree that these methods are not used much in the literature. We have recently shown their value in interpreting coherent structures in turbulent flows for other applications and expanded the discussion in this work to improve the understanding.

**POD analysis: Patch 2 scenario is suddenly abandoned in the POD results section without justification. That the POD results are conclusive for the presence of von Karman street is not so clear from the figures.**

We have now included the patch canopy 2 case as well in figure 5. It is true that from visual cues, it is not very clear whether VKS exist - that is why we proceed to the subsequent analyses. This is stated more clearly now.

**MIC: large and small scales are poorly defined.**

They are now explained better as the Lorenz curve method is described in more detail.

**Virtual towers of 20Hz frequency are used to compare with field results, although field results are never discussed. Is there a reason why the authors decided to use a 30 minute period rather than stationary, less noisy results?**

Similar LES setup in PALM are used in other publications to compare against field campaigns. Here we do not have any field reference, so we do not compare.  However, turbulent statistics for the horizontal homogeneous case in figure 4 compares well with standard results in canopy turbulence such as Dias Junior 2015, Patton 2016, Shaw and Schumann, 1992. It is a standard practice in field measurements of atmospheric turbulence to use half hour statistics. We want to be consistent wit that practice so that if somebody collects data on similar configurations, they can compare.

---

## Author Comment (AC2) · 13 Mar 2018

**Response to reviewer 2**

**General comments:**
**In this work, it is investigated the presence of von Kármán (VK) vortices and Kelvin-Helmholtz (KH) instabilities in Large-Eddy Simulation (LES) when the common distributed drag force approach is used to represent plant canopies. Three different simulations were performed: one with horizontally homogeneous drag force, and two with a different set of patched areas with drag force. Four different types of data analyses were used to characterize the presence of coherent structures in the LES flow field, namely proper orthogonal decomposition, Shannon entropy, mutual information content and synchronization analysis. In these analyses, several features were pointed out as indicators of the presence of coherent structures. It is concluded that a weak signature of these coherent structures is present in the drag force formulation of plant canopies.**
**Overall, the motivation of the work is interesting (how realistic the drag force formulation of canopy in LES is), but this work presented a superficial and unconvincing evaluation of the signature of coherent structures in the simulation. Also, it is not provided any comparison of the LES results with data from a real canopy, which in my view is highly needed. I would like to point out some specific issues and questions that could help to improve the work.**

We thank the reviewer for the constructive comments. We have improved the manuscript based on Your suggestions.

**Specific comments:**

1. **Abstract: it is already clear from the abstract that the work has more motivation than actual results or conclusions. After improving the content of the manuscript, the abstract should focus more on the results and conclusions of the work.**

Thanks for pointing out. We have modified the abstract accordingly.

**2. Introduction: a good description of the canopy layers, coherent structures and drag formulation is given, but the review on the current state of the art is poor. Paragraph between the lines 18 of page 2 and 5 of page 3 is a clear example. How do all of these works have contributed to the understanding of the canonical structure of homogeneous canopy? What did they find, and what didn't they find? Which gaps are these? More specifically, what kind of information regarding the signature of coherent structures in the canopy did they find, that could help to interpret your results? Is this the very first work testing how realist the drag force formulation is? Given how many times it has been used, I highly doubt that, but if this is the case, you should state it clearly.**

We have expanded this section according to these recommendations. In our opinion, this is the first work testing the distributed drag formulation in numerical simulations. Other numerical works use this formulation and takes the KH and VK structures as granted as they are shown by experiments involving real canopy obstructions. We have improved the discussion to reflect these.

**3. Methodology: I am not convinced by this "weak and more generic" or "weak and loose" definitions of VK and KH motions. What are the basis for them? Have they been**

**used as surrogate for coherent structures before? I believe there will be an enhanced motion in these directions due only to the difference in the flow caused by the drag, even if there is no coherent structure (which should be captured by the drag formulation). I expect, for example, that this weak signature may represent purely local fluxes that has little to do with coherent structures. Is there a way to differentiate purely local flux to the true coherent structure signature? I believe this is a difficult task, but if you are claiming that there is some weak signature of coherent structures in the drag formulation, I believe you should be able to show this better.**

It is a widely accepted fact in atmospheric and engineering applications that velocity inflections are associated with KH type instabilities. From literature review it appeared to us that the presence of KHI is somewhat taken for granted in for canopy turbulence. For example, the well-known review paper Finnigan 2000 summarizes results from Raupach 1996 which calculated mean streamwise separation of coherent eddies near canopy top and presented a conceptual model which looks like figure 1.

[Figure]

Figure 1: Conceptual diagram of KHI as shown in Finnigan 2000.

They also conclude that although these structures evolve throughout the stages of turbulence development, the streamwise wavelengths are preserved. This motivates our current work in the sense that we want to confirm the presence of such structures directly from the velocity signals instead of the more abstract turbulence statistics that are usually found in the literature.
       Similarly for the VK structures, it is an accepted fact that they evolve in a horizontal place across the flow behind the obstacle and for canopy turbulence, this is taken for granted although it is only the lab experiments with real obstacles that observe these. We wanted to examine how realistic are these assumptions for the drag formulation. The so called weak definitions are thus not found in the literature, and we construct them in the paper. What we mean is we want a more direct evidence from the velocity records themselves. We describe this in more clearly and also remove the terms 'weak definition' to avoid confusion.

**4. Results: I'm very confused about how it is possible to have VK vortices in the horizontally homogeneous case. What is the mechanism of creation of them, if there is no discontinuity in the horizontal plane? How do you know these are not just "regular"**

**turbulent motion (not created by coherent structures), but instead these are coherent structures? I would expect no VK vortices in the horizontally homogeneous case, and maybe something similar to VK vortices in the patches case due to the discontinuity (even though I believe in reality they are created by the complete blocking of the flow by plant trunks, and may not form from the reduced flow by a patch with drag). The KH vortices, yes, they are created by the inflection point which is present in the drag formulation, so they can be present in all cases (although in different ways based on the intensity of the inflection point). In order to differentiate "regular" turbulent motion from coherent structures, maybe a simulation without canopy could be used. Those snapshots of Figure 3 do not show me anything. I cannot even see what is different inside the squares with the "VK?" from the rest of the plot. And if it is not conclusive, why show it?**

Thanks for this comment. We were motivated by these exact same question - that for horizontally homogeneous canopies there are no discontinuity, although the VK motions are still used in the conceptual model. That is why we did the patch configurations to provide the discontinuity and variations of wake spaces to see if they produce a clearer signature of VK motions. That is why we looked for even weak signatures of u-v covariations as visually we could not find any strong signature as one would see in real experiments. This is explained better in the text. In simulations without canopy no such signature are found and we have included an appendix with results from a simulation without any canopy.

**5. Overall, in order to argue that any feature is due to the canopy presence, it is good to compare it with the case without canopy. Also, it is highly needed a direct comparison of LES results with cases in the literature where the coherent structures were really present. For example, you should compare LES statistics with results from wind tunnel or field data where a real canopy was present, and point out which features are present in the LES and which are not. Only after that, you could show these new analyses that (I believe) nobody has done before, and show how they improve the interpretation of your results. So far I have no way of telling what are you actually showing with these new analyses, and if the features that you found are actually any weak signature of coherent structures or something else. I think that you reached the conclusion that those features are from coherent structures too fast and without enough evidence.**

We have included an appendix with results from a no-canopy run and a section using data from a real canopy. The time and space averaged turbulent statistics as seen in figure 4 are already comparable to existing literature.

**Conclusions: if there are significant differences between the drag model and the real case, it should be discussed what are the implications of these differences for all the studies that are done with LES**.

Agreed. We have expanded the discussion in this direction.

---

## Author Comment (AC3) · 13 Mar 2018

**Response to reviewer 3**

**General Comments:**
**The paper evaluates the performance of a canopy model using a drag formula in the large eddy simulation (LES). Particularly, the existence of two key features –Kelvin Helmholtz (KH) waves and von Karman (VK) vortices – over the three canopy configurations are examined in this work. Several innovative approaches (proper orthogonal decomposition, Shannon entropy, Mutual information and synchronization analysis) are introduced to analyze the LES data. The conclusion confirms the existence of both phenomena in canopy flow simulations. However, overall, the analyses based on new approaches are incomplete and lack physical interpretation.**

We thank the reviewer for the constructive comments and suggestions. We have improved the manuscript based on these recommendations.

**Specific Comments:**
**Major (The authors need to well address these comments before the paper can be published):**

1. **The paper lacks some important information about the description of analysis approach. These include the lack of (i) the physical interpretation of mutual information content (MIC), (ii) the mathematical equation and physical meaning of synchronization analysis, and (iii) the approach to split the large scale and small scale in MIC analysis.**

This is pointed out by reviewer 1 as well. We have added discussions on all of them in the text and appendix.

2. **The description of LES setup is incomplete. For example, the paper doesn't pro- vide some key parameters of two patched configurations (e.g. the separation between patches) and the reason to select those key parameters. The reasons to choose different locations of virtual towers between two patched configurations are not given – Why the number of virtual towers inside selected canopy patches are different between these two configurations? The detailed description of initial conditions is missing.**

This was also pointed out by reviewer 1 and we have added discussions on all of these points.

3. **The use of virtual tower to mimic observations in lab or field experiment is interesting, but weakened by the lack of comparison with data from observations. This makes the conclusions much weaker, especially due to the use of innovative analysis approaches without any validation. The use of those approaches need to answer the following questions before being applied in a such case: can KH waves and VK vortices be interpreted using these approaches in a real case? and if they can, whether these phenomena can be observed in LES results?**

This is a great point and also pointed out by reviewer 2. We have added appendices and discussions to compare with no -canopy simulations and real canopy data.

**4. The KH and VK phenomena can be identified from many traditional approaches including quadrant analysis, skewness analysis, variable-interval time averaging, wavelet transform, two-point space-time correlation tensor, etc. (a good review about these approaches is in Finnigan (2000)). However, these methods are not being used in cur- rent analysis. Why? For example, the following questions is critical: can KH and VK be identified using traditional approaches in current simulations? compared to these traditional approaches, what are the advantages of using these innovative methods? If there are no advantages a a, why bother?**

It is a widely accepted fact in atmospheric and engineering applications that velocity inflections are associated with KH type instabilities. From literature review it appeared to us that the presence of KHI is somewhat taken for granted in for canopy turbulence. For example, the well-known review paper Fininigan 2000 summarizes results from Raupach 1996 which calculated mean streamwise separation of coherent eddies near canopy top and presented a conceptual model which looks like figure 1

[Figure]

Figure 1: Conceptual diagram of KHI as shown in Finnigan 2000.

They also conclude that although these structures evolve throughout the stages of turbulence development, the streamwise wavelengths are preserved. This motivates our current work in the sense that we want to confirm the presence of such structures directly from the velocity signals instead of the more abstract turbulence statistics that are usually found in the literature.

Similarly for the VK structures, it is an accepted fact that they evolve in a horizontal place across the flow behind the obstacle and for canopy turbulence, this is taken for granted although it is only the lab experiments with real obstacles that observe these. We wanted to examine how realistic are these assumptions for the drag formulation. The so called weak definitions are thus not found in the literature, and we construct them in the paper. What we mean is we want a more direct evidence from the velocity records themselves. We describe this in more clearly and also remove the terms 'weak definition' to avoid confusion.

**5. The turbulent flow should be more or less symmetric with respect to the central plane of each canopy patch in streamwise direction. Thus, the turbulent flux  should be**

**more or less zero at the virtual towers located on these planes. However, in Figure 4, the turbulent flux  is non-zero in all three cases. This phenomenon is not explained.**

This is because we do not employ any spatial averaging. This discussion is added in the text.

**6. Some panels of figures are not analyzed in current manuscript and should not appear in the paper. These panels include the length scale of u in Figure 4 and third column of Figure 5.**

Removed.

**Minor:**
1.  **The resolution of Figure 3 and Figure 5 are too low in current manuscript which make them difficult to view. Meanwhile, the ranges of coordinates and color bar in both figures are not the same from case to case.**

Improved.
**2. The title of section 4.3 is already included in title of section 4.2 and need to be corrected.**

Corrected.
**3. The index i shouldn't be used repetitively in equation 'pi=. . .' in page 7 line 7.**

Corrected.

**4. New variable V in page 11 line 8 need to be defined before being used.**

Corrected.

**5. Some other text typos and format problem associate with units need to be corrected.**

Checked and corrected.